# Prevalence of Acute Hepatitis E Virus Infections in Swiss Blood Donors 2018–2020

**DOI:** 10.3390/v16050744

**Published:** 2024-05-08

**Authors:** Christoph Niederhauser, Peter Gowland, Nadja Widmer, Soraya Amar EL Dusouqui, Maja Mattle-Greminger, Jochen Gottschalk, Beat M. Frey

**Affiliations:** 1Interregional Blood Transfusion SRC, 3008 Berne, Switzerland; peter.gowland@itransfusion.ch (P.G.);; 2Institute of Infectious Disease, University of Berne, 3008 Berne, Switzerland; 3National Blood Transfusion Service of the Swiss Red Cross, 3008 Berne, Switzerland; soraya.amar@blutspende.ch; 4Regional Blood Transfusion SRC, 8952 Schlieren, Switzerland; m.mattle@zhbsd.ch (M.M.-G.); j.gottschalk@zhbsd.ch (J.G.); bm.frey@zhbsd.ch (B.M.F.)

**Keywords:** blood donors, universal blood donor screening genotype, hepatitis E virus phylogenetic analysis, Switzerland, surveillance, zoonotic infection

## Abstract

Introduction: Hepatitis E virus (HEV) genotype 3 is the major cause of acute viral hepatitis in several European countries. It is acquired mainly by ingesting contaminated pork, but has also been reported to be transmitted through blood transfusion. Although most HEV infections, including those via blood products, are usually self-limiting, they may become chronic in immunocompromised persons. It is thus essential to identify HEV-infected blood donations to prevent transmission to vulnerable recipients. Aims: Prior to the decision whether to introduce HEV RNA screening for all Swiss blood donations, a 2-year nationwide prevalence study was conducted. Methods: All blood donations were screened in pools of 12–24 samples at five regional blood donation services, and HEV RNA-positive pools were subsequently resolved to the individual donation index donation (X). The viral load, HEV IgG and IgM serology, and HEV genotype were determined. Follow-up investigations were conducted on future control donations (X + 1) and previous archived donations of the donor (X − 1) where available. Results: Between October 2018 and September 2020, 541,349 blood donations were screened and 125 confirmed positive donations were identified (prevalence 1:4331 donations). At the time of blood donation, the HEV RNA-positive individuals were symptom-free. The median viral load was 554 IU/mL (range: 2.01–2,500,000 IU/mL). Men (88; 70%) were more frequently infected than women (37; 30%), as compared with the sex distribution in the Swiss donor population (57% male/43% female, *p* < 0.01). Of the 106 genotyped cases (85%), all belonged to genotype 3. Two HEV sub-genotypes predominated; 3h3 (formerly 3s) and 3c. The remaining sub-genotypes are all known to circulate in Europe. Five 3ra genotypes were identified, this being a variant associated with rabbits. In total, 85 (68%) X donations were negative for HEV IgM and IgG. The remaining 40 (32%) were positive for HEV IgG and/or IgM, and consistent with an active infection. We found no markers of previous HEV in 87 of the 89 available and analyzed archive samples (X − 1). Two donors were HEV IgG-positive in the X − 1 donation suggesting insufficient immunity to prevent HEV reinfection. Time of collection of the 90 (72%) analyzed X + 1 donations varied between 2.9 and 101.9 weeks (median of 35 weeks) after X donation. As expected, none of those tested were positive for HEV RNA. Most donors (89; 99%) were positive for anti-HEV lgG/lgM (i.e., seroconversion). HEV lgM-positivity (23; 26%) indicates an often-long persistence of lgM antibodies post-HEV infection. Conclusion: The data collected during the first year of the study provided the basis for the decision to establish mandatory HEV RNA universal screening of all Swiss blood donations in minipools, a vital step in providing safer blood for all recipients, especially those who are immunosuppressed.

## 1. Introduction

Hepatitis E virus (HEV; family Hepeviridae, subfamily *Orthohepevirinae*, *genus Paslahepvirus*) is an RNA virus that causes acute viral hepatitis in humans [1]. Five HEV genotypes are known to infect humans. In Europe, most HEV cases are caused by the food-borne transmission of genotype 3 and transmitted via the consumption of raw and undercooked liver, meat, or sausages from domestic pigs. Several studies have detected HEV in porcine liver, pork and pork products, or improperly cooked pig meat. Transmission by direct contact with infected animals is also observed. Since the mid-2000s, parenteral transmission of HEV infections via contaminated blood products and organs has been documented [2,3,4]. Annually, greater than two million locally acquired HEV infections occur in Europe. An acute HEV infection is typically self-limiting and sub-clinical [5]. Nevertheless, HEV infection poses a significant risk to persons who have compromised immune systems because they can develop a persistent HEV infection, with rapid progression to cirrhosis, decompensation, and even death [6]. Moreover, HEV genotype 3 infection has also been recently associated with several extrahepatic manifestations, including Guillain–Barré syndrome, inflammatory polyradiculopathy, bilateral brachial neuritis, encephalitis, ataxia/proximal myopathy, and necrotizing myositis [6,7]. There is no proven treatment for chronic HEV infection, although ribavirin therapy or the reduction of immunosuppression have been successful in achieving HEV RNA clearance [8,9].

HEV infection emerged as a new threat to blood safety in the early 2000s, with the first case of transfusion transmission being reported in Japan in 2002 [10]. Since then, several cases of transfusion-transmitted HEV (TT-HEV) infections have been reported in various European countries, involving all types of blood components including solvent detergent (SD)-treated plasma and pathogen-reduced components [4,11,12,13,14,15,16,17]. In addition, it has been suggested that HEV infections might cause chronic hepatitis in immunosuppressed recipients [18,19].

A major systematic study assessing HEV transmission through blood transfusion was conducted in England during 2012–2013 [4]. This study reported that 42% of recipients who received HEV RNA-contaminated blood components showed evidence of infection, and a progression to a chronic infection was demonstrated in 50% of HEV-infected immunosuppressed recipients.

Due to an increasing number of TT-HEV cases and the severity of the disease in immunocompromised patients, several countries have chosen to implement the HEV-RNA screening of blood donations to mitigate the risk of transmission through blood components. Japan and a few European countries, such as the United Kingdom, Ireland, the Netherland, the Catalonian province of Spain, some blood banks in Germany, have implemented HEV RNA testing. Some have employed a universal screening approach, while others have adopted a selective screening procedure targeted towards specific at-risk recipients; still others have limited the testing to certain regions of the country [20,21,22,23]. Of the 27 EU member states, several have implemented NAT screening for HEV RNA. Universal donor screening is performed in several countries, including Ireland, France, the Netherlands, Switzerland, and UK. Selective screening, which assures the inventory of HEV-negative blood for high-risk patients, is conducted in Austria and Luxembourg. Some countries perform the universal screening of blood donations collected in specific, high-prevalence regions, e.g., Catalonia and Asturias in Spain, and North Rhine–Westphalia and Hamburg in Germany [24]. After performing HEV seroprevalence studies, Sweden and Denmark decided not to introduce screening for HEV RNA [25,26].

In Switzerland, the National Blood Transfusion Service of the Swiss Red Cross (BTS SRC) decided in 2018 to introduce a general HEV RNA screening for all blood donations collected with a minimum sensitivity of 450 IU/mL in the individual donation. An important reason for that decision was given by the data collected during the seroprevalence study carried out in 2016 on almost 4000 blood donors in Switzerland [27] and, of course, the data compiled in similar studies throughout Europe. The BTS SRC initially proposed a 2-year surveillance period of the HEV RNA screening, during which data were to be collected and evaluated. After this period, the analyzed data were used to provide the basis for a definitive decision on whether to continue with a mandatory universal HEV RNA screening of all Swiss blood donations. Here, we report the results of this 2-year screening of 541,349 Swiss blood donations for HEV RNA between October 2018 and September 2020. We aimed to determine the incidence of HEV among Swiss blood donors, and to measure the respective viral loads of these donations, in order to identify the circulating HEV genotypes and sub-genotypes and to determine the timing of HEV seroconversions among the blood donors.

## 2. Materials and Methods

### 2.1. Study Population, HEV RNA Detection and Characterization

All whole-blood and apheresis donations collected between 1 October 2018 and 30 September 2020 by the regional Swiss blood transfusion services were tested for HEV RNA (Figure 1). The number of HEV-positive donations was used to calculate HEV’s incidence in the Swiss donor population. During this period, 541,349 blood and apheresis donations were collected from the whole of Switzerland and sent to one of five regional blood donation test centers to be analyzed for HEV RNA. At these centers, minipools of 12–24 samples were constructed and two commercial HEV RNA assays (cobas HEV for use on the 6800/8800 platforms (Roche Diagnostics, Rotkreuz, Switzerland) and Procleix HEV assay on the Procleix Panther platform (Grifols Diagnostic Solutions, Emeryville, CA, USA)) were used for the screening. The reason for the different assays and pool sizes used in Switzerland is the federal system of 11 independent BTS. The only prerequisite is that the sensitivity limit of 450 IU/mL in the individual donation must be achieved. The cobas HEV RNA test is an internally controlled RT-PCR with a reported 95% limit of detection of 18.6 IU/mL, whereas the Procleix HEV assay is a transcription-mediated amplification (TMA) assay with a reported 95% limit of detection of 7.9 IU/mL. With the cobas HEV assay mini-pools of 16 to 24 donations and with the Procleix HEV test mini-pools of 12 to 15 were analyzed. The sensitivity limits for the individual donations with the assays used and the corresponding minipools were as follows: cobas HEV with pools of 16 donations—298 IU/mL, and with pools of 24 donations—446 IU/mL, and for the Procleix HEV assay in pools of 12 donations—96 IU/mL, and in pools of 15 donations—120 IU/mL. Reactive pools were subsequently resolved to the individual HEV RNA-containing donation (X index donation) with the same assay as was used for the pool screening. All the individual X-donations were subjected to further confirmation testing at the Swiss blood donor reference laboratory. The viral load was determined with the RealStar HEV RT-PCR assay (Altona Diagnostics GmbH, Hamburg, Germany). The in-house-determined lower limit of detection (95% cut-off) of this assay was 5.0 IU/mL, conducted using a calibration curve based on the WHO international standard for HEV RNA (PEI code 6329/10; https://www.pei.de; accessed on 1 March 2024). HEV serology was performed using the Wantai HEV IgG and IgM detection assays according to the manufacturer’s instructions (Eurobio, Les Ulis, France).

### 2.2. Statistical Analysis

The Clopper–Pearson exact method was used to calculate the 95 percent confidence intervals of HEV-RNA prevalence.

### 2.3. Phylogenetic Analysis

Viral RNA from 800 µL plasma was extracted using the Qiagen Virus BioRobot 9604 kit (Qiagen, Hombrechtikon, Switzerland) modified for a Tecan Genesis platform (Tecan AG, Männedorf, Switzerland) and eluted in 80 µL Qiagen elution buffer AVE. To determine the HEV genotype and sub-genotype of positive samples, a broad reactive nested RT-PCR was performed [28]. Viral RNA was converted to cDNA using random hexamers and Superscript IV enzyme (Thermo Fischer Scientific, Reinach, Switzerland) according to the manufacturer’s instructions, followed by a nested typing RT-PCR using HotStarTaq DNA Polymerase (Qiagen, Hombrechtikon, Switzerland), creating a 493-nucleotide product of the 5′ region of the ORF2 from the HEV genome (position 6029 to 6521 of reference genome AB630970). An alternative RT-PCR was conducted on some templates producing, a 589-nucleotide ORF2 product [29] (position 5752 to 6340 of reference genome AB630970). The forward and reverse primers were used for bi-directional sanger sequencing (Mycrosynth GmbH, Balgach, Switzerland). To determine the HEV genotype and sub-genotype, the sequences were submitted to the online HEVnet typing tool (https://www.rivm.nl/mpf/typingtool/hev/; accessed on 13 June 2023) and a 312-nucleotide overlapping sequence position (position 6029 to 6340 of reference genome AB630970) was phylogenetically analyzed using HEV reference strains [30,31]. Sequences were aligned using MUSCLE, and a Maximum Likelihood Tree with 1000 bootstrap replicates was created using the Tamura–Nei model [32] (MEGA X software, version 10.1.7). The sequences have been submitted to the Genbank database, and the accession numbers are OV844702-OV844807 (Appendix A).

### 2.4. Archived Samples and Follow-Up Donations

The archived plasma samples (X − 1) of the most recent donations were retrieved when available for all RNA-positive donations and tested for HEV RNA and antibodies. If this donation was contaminated with HEV RNA, then the next most recent donation (X − 2) was retrieved and tested. Hospitals that had received the HEV-containing blood products were informed, and a look-back procedure was conducted. All available follow-up donations (X + 1) were investigated for HEV RNA and serology (Figure 1).

## 3. Results

Between 1 October 2018 and 30 September 2020, a total of 541,349 Swiss whole blood and apheresis blood donations were screened for HEV RNA, and 125 donations were identified as positive (Figure 1, Table 1), showing an incidence of 1:4331 (95% confidence interval (CI) 1:3635–1:5203) donations (0.023%). Four initial positive screening samples were not confirmed with the quantitative HEV PCR assay (Figure 1). From two cases, a follow-up sample several months later was received, and they were both HEV IgG- and IgM-negative. It is thus assumed that these four samples were false-positive. The reason for these false positive results was not identified. There was not a considerable difference observed in the number of cases identified using the different screening strategies (i.e., HEV RNA assay and mini-pool size, Appendix A). At the time of blood donation, the HEV RNA-positive individuals were symptom-free. On average, 5.2 ± 2.9 HEV RNA-positive donations were identified per month (Figure 2), with a possible decrease in this rate over the 2-year study period. The median viral load was 554 IU/mL (Table 1). A broad range of viral loads was measured in the positive donations (2.01 to 2,500,000 IU/mL), with greater than 50% below 1000 IU/mL (57%; Table 1 and Table 2). Male donors (88; 70%) were significantly more frequently infected than female donors (37; 30%), as compared to the sex distribution in the Swiss blood donor population (57% male/43% female, *p* < 0.01) (https://www.blutspende.ch/; accessed on 1 March 2024). From the donations extrapolated from male and female donors (males: 308,569 and females 232,780), the HEV prevalence in these donors was calculated: male donors 1:3506 (95% confidence interval (CI) 1:2846–1:4372) compared to 1:6261 (95% confidence interval (CI) 1:4565–1:8935) in female donors (Table 1). Of the 125 samples, 106 were successfully genotyped (85%); all belonged to genotype 3 (Table 3, Figure 3). Most of the virus strains clustered with two distinct HEV sub-genotypes; 3h3 (formerly 3h-s (p)), a variant endemic in the Swiss pig population and which is practically confined to Switzerland, and 3c, a variant often encountered in northern Europe. The remaining sub-genotypes identified are all known to circulate in Europe. Five 3ra sub-genotypes were identified, a variant associated with rabbits (Table 3, Figure 3, and Appendix A and Appendix A). More than two thirds of the samples from the HEV-positive donors (85/125, 68%) were window period infections, as they were unreactive for anti-HEV IgG and anti-HEV IgM at the point of donation, whereas 25 donors (20%) were reactive for both anti-HEV IgG and anti-HEV IgM, 10 (8%) were reactive only for anti-HEV IgG antibodies, and finally the remaining 5 (4%) were reactive for anti-HEV IgM antibodies alone (Table 4A).

A previously negative or not tested archive (X − 1) plasma sample was available for 87 (70%) of the 125 identified HEV RNA-positive index donations (Table 4B). A further 18 donors were first-time donors, thus no previous plasma was available to analyze; 2 of these plasma samples were found to be weakly HEV RNA-positive (result of HEV IgG assay, S/CO = 1.1). They were both derived from platelet apheresis donations and had viral loads of 35.0 IU/mL and 6.3 IU/mL, which are both clearly under the 95% limit of detection of the screening HEV RNA assay used in the mini-pools, and were taken 4 and 10 weeks prior to the index X donation, respectively. Both were HEV IgG- and IgM-negative, and both were HEV IgG-reactive in the X donation. No HEV RNA was found in the next most recently archived plasma sample of these donors (X − 2). Two donors were weakly HEV IgG-positive in the X − 1 donation, suggesting insufficient immunity to prevent an HEV reinfection.

Follow-up samples (X + 1) were obtained from 90 (72%) donors (Table 4C). These samples were collected between 2.9 and 101.9 weeks after the index (X) donation (median 35 weeks). As expected, none of those tested were positive for HEV RNA. In all cases except one (89, 99%), the donor developed an HEV anti-IgG and/or anti-IgM response. Persistent HEV IgM antibodies were detected in 23 of the follow-up X + 1 samples, collected up to 20 months (median 8 months) after the index X donation. The single donor that did not develop HEV anti-IgG or anti-IgM antibodies was tested in samples taken 90 (X + 1) and 105 (X + 2) weeks after the index (X) donation (viral load 221 IU/mL, subtype HEV 3c).

## 4. Discussion

In Switzerland, the universal screening of blood donations for HEV RNA was only introduced in 2018, although HEV seroprevalence in the blood donor population averaged 20% in a study conducted between 2014 and 2016 [27]. This study also showed that HEV seroprevalence increased with the age of the donors, suggesting that HEV is constantly circulating in the Swiss population. This study thus formed the basis for the current HEV RNA blood donor screening study, with the aim of determining the HEV RNA incidence in blood donors and thus reducing the risk of TT-chronic HEV infections in immunocompromised patients. During the two-year study period, which began in October 2018, a high incidence of HEV infection was detected in the collected blood. HEV RNA was confirmed in 125 of 541,349 blood donations, corresponding to an overall incidence of 1 in 4331 donations (0.023%). All these asymptomatic but HEV RNA-positive donations were successfully eliminated from the Swiss blood supply. A similar prevalence of viraemia among blood donors has previously been reported in many other European countries, ranging from 1 in 744 to 1 in 8636 donations [21,22,23,33,34,35,36,37,38,39,40,41]. HEV RNA screening was not introduced in all European countries despite the measured HEV RNA prevalence among blood donors in the individual countries. Sweden, a country with low HEV RNA prevalence, decided against introducing HEV NAT screening due to its low prevalence. On the other side, countries with higher HEV RNA prevalence, such as Poland and Denmark, have also decided against mandatory nationwide screening [25,42]. Our results suggest that Switzerland is in an area of intermediate endemicity that requires universal HEV RNA screening.

During the two-year study period, HEV-infected donors were evenly distributed across Switzerland, and no outbreaks were observed. The average age of HEV-infected donors was higher in male donors than in female donors (47.5 years versus 38.4 years). The incidence of HEV was higher in male blood donors than in female donors (70% versus 30%). This gender difference has been repeatedly observed in previous studies of blood donor populations across Europe [23,40,43,44]. One explanation for the predominance of male cases could be a difference in meat consumption [45]. An alternative explanation for the gender difference is that men are more clinically susceptible due to gender differences, while women are less likely to have acute clinical disease [45]. The reason why more HEV cases are male remains unclear and requires further research. In contrast to reports from other countries, the HEV RNA prevalence observed in this study remained stable in younger and older donors [21,23,40].

Despite the success of HEV RNA screening, several studies and case reports have suggested that HEV transfusion-transmitted infections can still result from donations with low viral loads (VL) [46]. The limit of detection of the HEV screening strategy is inversely dependant on the NAT approach used (i.e., MP- or ID-NAT). Currently, there is no agreed standard for choosing between ID and MP-NAT strategies, so practices in screening laboratories in Europe and worldwide vary widely. While the sensitivity of ID-NAT is obviously higher than that of MP-NAT, the implementation and organizational options often make the ID-NAT approach prohibitively expensive for large-scale HEV RNA screening. Regardless of the inherent trade-off between cost and sensitivity, the main problem in selecting a screening protocol is that there are no definitive data on the minimum VL at which TT-HEV infection can occur [4,21]. Furthermore, it is the infectious dose that is relevant to the risk of transmission, and different blood products invariably feature large differences in plasma content [19]. For example, products with lower plasma volume components (e.g., red blood cells and platelets) are less likely to transmit HEV than products with high plasma volume components (e.g., fresh frozen plasma). According to one report, the average infectious dose that leads to HEV infection is 520,000 IU, regardless of the recipient’s immune status. In contrast, Tedder et al. were unable to detect TT-HEV infection in recipients of donor samples with <19,000 IU. The minimum VL most likely lies between these two extremes, with the actual value being closer to the estimates of the latter [47]. However, perhaps the biggest hurdle to achieving a reliable minimum VL for TT-HEV is that the quantification of HEV RNA levels depends on the detection limit of the analytical system used. The MP-NAT limits of detection (LOD) in two recently published studies were around 450 IU/mL [33,48]. Although this sensitivity is generally impressive, it may be insufficient for routine blood bank testing, as detectable HEV viraemia in asymptomatic blood donor samples is often significantly lower than in serum samples from patients with acute infection. For example, several reports of HEV ID-NAT blood donor testing from Ireland showed that 59% of HEV-responsive samples had a VL of < 450 IU/mL [21,22,49,50], which would have placed them below the detection limit of the Grifols HEV RT-PCR and Cobas HEV tests, and thus they would have been categorized as false-negative. Indeed, our data show a similar relatively low median level for VL detected in our 125 HEV RNA-positive blood donations (554 IU/mL), with 57% of VL below 1000 IU/mL and 20% even below 100 IU/mL. This compares well with the VL measured in Swiss patients diagnosed with acute hepatitis E infection [51]. It can be assumed that the 12–24 pool MP-NAT approach used in Switzerland with a 95% LOD between 96 and 446 IU/mL may have overlooked donations with very low VL. Whether such cases are infectious has recently been the subject of intense debate among experts, with the conclusion being reached that although the risk of TT-infections is relatively low in immunocompetent individuals, it can be a problem in immunocompromised patients. Despite these reports of low VL in the infected blood products, these products still occasionally lead to TT-HEV, as reported in a case from Germany. An immunocompetent patient received an apheresis platelet transfusion from an HEV-infected single donor (120 IU/mL HEV RNA/mL plasma having an infectious dose of 8892 IU) [16]. Similar cases have been reported in other countries [52,53]. In a similar vein, HEV transmission due to low HEV RNA concentrations in contaminated organ donations has recently been documented [54]. These cases suggest that we still have much to learn regarding recipient factors leading to chronic HEV infection, and suggest that a re-evaluation of the VL threshold for TT-HEV in blood products may be necessary for ensuring transfusion safety and treating overall disease.

None of the infected donors showed signs of acute or chronic HEV symptoms. Overall, 68% of HEV RNA-positive donors were seronegative at the time of donation and were thought to be in the early stages of infection. Some were reactive for only anti-HEV IgG antibodies without IgM reflecting late infections. Previously available archival donations (X – 1) were 96% seronegative, suggesting that most HEV RNA donors had no previous HEV contact, and thus reinfection was considered unlikely. All but one of the follow-up donations (99%, X + 1) showed seroconversion for HEV IgG and/or IgM. It remains unclear why one donor did not develop HEV antibodies despite being infected with HEV.

The HEV NAT screening of blood donors will reduce the risk of HEV transmissions, as well as cases with chronic disease. Depending on test sensitivity and NAT pool size (pool of 96 to ID-NAT testing), a reduction of 80% down to 99% can be achieved in transmissions and subsequent chronic cases [55]. These risks can be calculated using complex algorithms employing detection rates observed with the ID or MP-NAT approaches, as has been done for the three most important viruses (i.e., HIV, HCV, and HBV in combination). The risk for HEV is considerably higher than for these other viruses [56]. In Switzerland, the risk assessments considered both HEV seroprevalence and HEV RNA incidence, and the results led to a decision to introduce the mandatory HEV RNA screening of all Swiss blood donations in summer 2022.

Our study also shows the diversity and uniqueness of the HEV strains circulating in Switzerland. A partial ORF-2 region DNA sequence was obtained from 85% of the HEV-positive donations (106 from 125). Phylogenetic analysis revealed that all belonged to genotype 3 (Figure 3, Table 3, Appendix A and Appendix A). Three subtypes, 3c (40, 38%), 3h3 (42, 40%) within clade-2/abchijklm, and 3f1 (10, 9%) within clade-1/efg, were dominant. Variant 3h3 is the most common subtype circulating in Swiss pigs and wild boars, as well as in occasional contaminated meat products [40,57,58,59]. It appears to be confined to Switzerland due to the high degree of self-sufficiency in Swiss pork consumption [57]. Variants 3c and 3f1 often predominant in other neighboring European countries, but are not often encountered in Swiss pigs and wild boar, as the import of live animals is strictly limited [40,58,60]. The high incidence of these variants is thus assumed to be due to the consumption of contaminated imported food products or infection while eating in neighboring countries [57]. The HEV RNA concentrations measured in those cases that were subtyped did not show a significant difference (Appendix A). Of particular interest is the relatively high incidence of the unique HEV variant 3ra found in our cohort of cases (5, 5%). This variant is predominantly detected in rabbits, but has occasionally been found in humans [61,62,63]. Whether the cases are a consequence of direct contact with rabbits or were due to foodborne or waterborne infections could not be determined.

## 5. Conclusions

Research and surveillance studies, expert groups, institutional bodies and regulators have documented the emergence of HEV, and demonstrated it to be a relatively frequent transfusion-transmissible agent in European blood donors. HEV is present not only in the food we consume, but also in our environment [64]. The response by blood services to HEV risk is influenced by epidemiological, economic, and political factors. In some countries, blood screening for HEV RNA has not been considered necessary at this time. Universal HEV ID-NAT testing almost completely excludes donations from asymptomatic viremic donors, but is rather costly. Switzerland, like many European countries, opted to implement an MP-NAT screening approach despite universal pathogen reduction strategies for platelet concentrates and the use of fresh frozen plasma to reduce transmission risk [65]. The elimination of HEV in the population will, however, require a “one health” approach, whereby the interrelationships between animals, food, human infection, and the environment are elucidated, and the optimal control points are then identified [66].

## Figures and Tables

**Figure 1 viruses-16-00744-f001:**
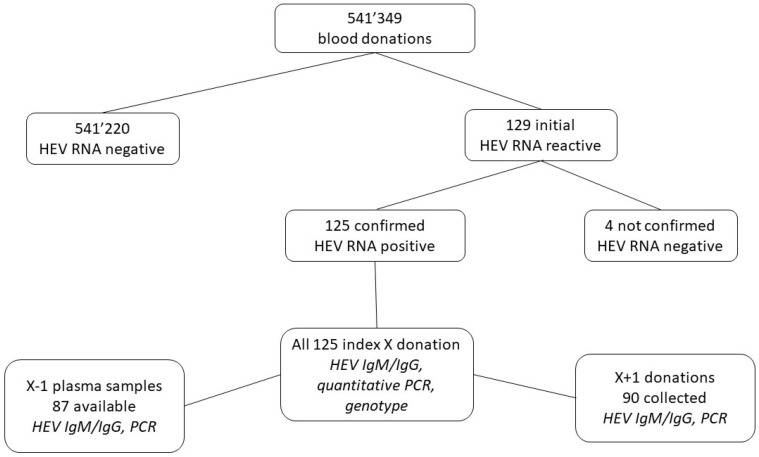
Overview of the proceedings for the described study.

**Figure 2 viruses-16-00744-f002:**
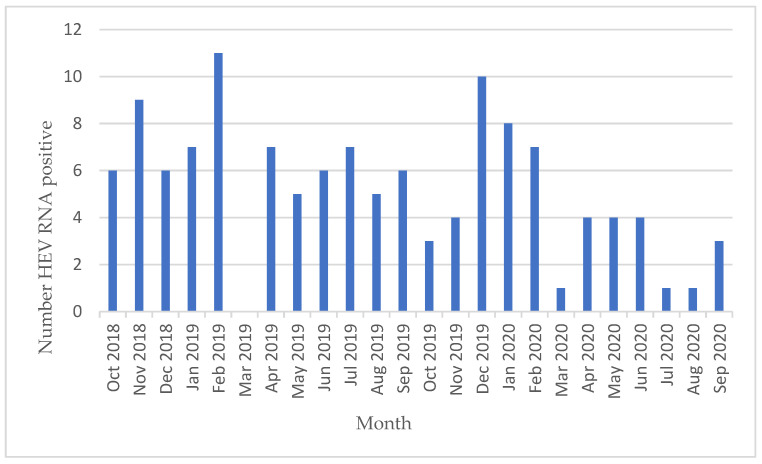
Confirmed HEV-positive cases per month from 1 October 2019 up to 30 September 2020.

**Figure 3 viruses-16-00744-f003:**
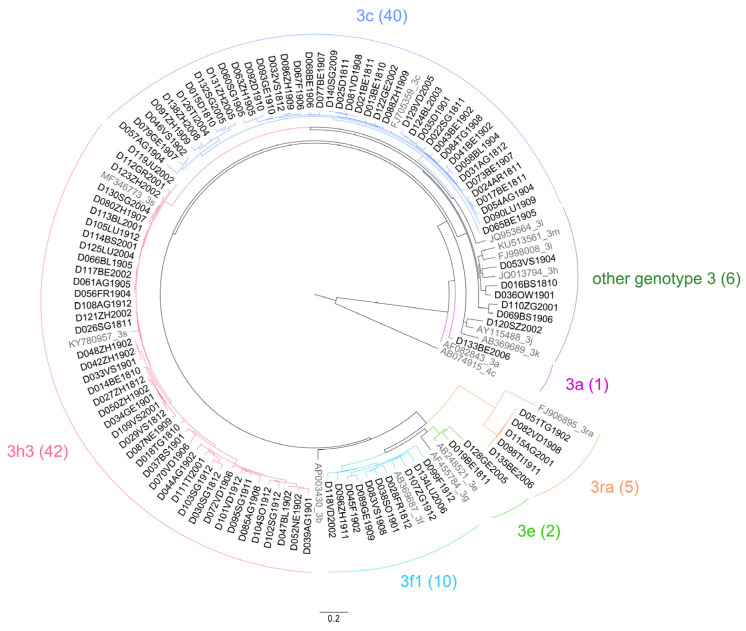
Circular phylogenetic analysis of the HEV variants from Swiss blood donors, October 2018–September 2020.

**Table 1 viruses-16-00744-t001:** Characteristics of the HEV RNA-positive blood donations.

Number donations screened	541,349	males: 308,569 females: 232,780
Number HEV RNA confirmed positive	125	males: 88 (70%) females: 37 (30%)
HEV prevalence (95% CI)	1:4331 donations (95% CI, 1:3635–1:5203)	males: 1:3506 (95% CI, 1:2846–1:4372) females: 1:6261 (95% CI, 1:4565–1:8935)
HEV median viral load	554 IU/mL	males: 734 IU/mL females: 266 IU/mL
HEV viral load range IU/mL	2.01–2,500,000 IU/mL	males: 4.15–2,500,000 IU/mL females: 2.01–85,802 IU/mL
Median age (y)	46.3 y	males: 47.5 y females: 38.4 y

**Table 2 viruses-16-00744-t002:** Distribution of viral loads in the HEV RNA positive donations.

HEV Viral Load IU/mL	Number Donations	Percentage (%)
1–10	7	
11–100	18	57%
101–1000	46	
1001–10,000	35	
10,001–100,000	10	43%
100,001–1,000,000	6	
1,000,001–10,000,000	3	
Total	125	

**Table 3 viruses-16-00744-t003:** Distribution of HEV sub-genotype 3 identified.

HEV Sub-Genotypes	Number (%)
HEV 3h3 (formerly 3h-s)	42 (40)
HEV 3c	40 (38)
HEV 3f1 (formerly 3f)	10 (9)
HEV 3ra	5 (5)
HEV 3e	2 (2)
HEV 3a	1 (1)
various HEV 3 (3/3l (formerly 3o(p)/3t(p) (formerly 3 unclassified))	6 (6)
Total	106

**Table 4 viruses-16-00744-t004:** Analysis of: A. Index (X). B. Archive (X − 1). C. Follow-up samples (X + 1).

**A. Serology (X) Index (X) donations**	
Number X index donations	125
HEV IgG- and IgM-negative (%)	85 (68%)
HEV IgG- and IgM-positive (%)	25 (20%)
HEV IgG only (%) ^1^	10 (8%)
HEV IgM only (%)	5 (4%)
**B. Characteristics X − 1 archive donations**	
Number X − 1 donations	87 (70%)
Number HEV RNA-positive	2 (2.3%)
Median time between X − 1 and X	147 days
HEV IgG- and IgM-negative (%)	85 (98%)
HEV IgG- and IgM-positive (%)	None
HEV IgG only (%)	2 (2.3%)
HEV IgM only (%)	None
**C. Characteristics X + 1 follow-up donations**	
Number X + 1 donations	90 (72%)
Number HEV RNA positive	None
Median time between X and X + 1	35 weeks (range 2.9–101.9 weeks)
HEV IgG- and IgM-negative (%)	1 (1.1%)
HEV IgG- and IgM-positive (%)	22 (24%)
HEV IgG only (%)	66 (73%)
HEV IgM only (%) ^2^	1 (1.1%)
**Summary X + 1 follow-up donations**	
HEV IgM-positive (%)	23 (26%)
HEV IgG-positive (%)	88 (98%)
HEV IgG- and/or IgM-positive (%)	89 (99%)

^1^ 5 of the 10 had very high IgG concentrations (S/CO > 15), while the remaining 5 had S/CO values between 1.1 and 3.5. ^2^ The X + 1 IgM-positive sample was IgG-negative, but had an S/CO value 0.8, slightly below the cut-off of 1. The IgM value was S/CO 1.6. It was collected 7 months after the index donation (X).

## Data Availability

Data are contained within the article and Appendix A. Further data will be shared upon request to the authors.

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
