# Peer review of "Prevalence of Acute Hepatitis E Virus Infections in Swiss Blood Donors 2018–2020"

_viruses, 2024, doi:10.3390/v16050744_

Round 1

Reviewer 1 Report

Comments and Suggestions for Authors

This manuscript reports the data of 2-year HEV RNA screening for all Swiss blood donations. This topic is important not only for improving safety of blood transfusion but also to better understand the spread of HEV in Switzerland as in other countries in Europe.

The manuscript is clearly written but there are several limitations.

Comments

1.       The 2-year HEV RNA screening was introduced on 1 October 2018 and ended on 30 September 2020 (Methods and Results line 179). What about after 30 September 2020 ? In addition, it is mentioned that the decision for the introduction of mandatory HEV RNA screening of all Swiss blood donations was taken in summer 2022 (Discussion line 361). Due to the submission date (March 2024), the data should be presented until the end of 2023. An update is needed. 

2.       The screening strategy was based on minipools of 12-24 samples (16-24 donations with the Roche Cobas assay and 12-15 with the Grifols Procleix assay). What is the rationale of these differences in the number of samples tested for each assay ?

3.       Due to several limits of detection for HEV RNA screening (limit of detection of each assay, size of the minipool) the number of positive samples and the number of donations tested should be indicated for each screening strategy. This is important for comparisons with other studies performed in Europe. These data are also relevant for a better estimation of HEV incidence in Switzerland.  

4.       Two donors were weak HEV IgG positive in the X-1 donation indicating a reinfection. What is the anti-HEV IgG concentration in WHO units/ml ?

5.       Discussion, lines 310-317 : these sentence should be rewritten because the key point is the infectious dose as discussed on line 341.

6.       Minor points

-          Introduction : ICTV classification has changed (Purdy J Gen Virol 2022). HEV is now Paslahepevirus balayani.

-          Introduction : an update of the screening strategy adopted for each country (universal or selective screening, individual testing or minipool testing/size of the pools) is needed. For instance, an universal screening has replaced a selective screening in France.

-          Discussion, line 294 : the predominance of male cases is not explained by more frequent acute clinical disease because HEV RNA screening is systematic in blood donors.

Author Response

HEV article 2024

Answers to reviewers’ comments.

Reviewer 1

  1. The 2-year HEV RNA screening was introduced on 1 October 2018 and ended on 30 September 2020 (Methods and Results line 179). What about after 30 September 2020? In addition, it is mentioned that the decision for the introduction of mandatory HEV RNA screening of all Swiss blood donations was taken in summer 2022 (Discussion line 361). Due to the submission date (March 2024), the data should be presented until the end of 2023. An update is needed.

This manuscript describes the 2-year HEV RNA screening project which began on 1st October 2018 and ended on 30th September 2020 and was initiated by the National Blood Transfusion Service of the Swiss Red Cross under the direction of the two largest regional blood transfusion services in Bern and Zurich. During this period, all HEV RNA positive index donations (X) were extensively investigated molecularly and serologically, and efforts were made to investigate all possible previous donations (X-1) and follow-up samples (X+1). It was a clearly defined project with a defined time limit set by the National Blood Transfusion Service that included the extra very specific investigations on these index (X), previous (X-1) and follow-up (X+1) donations. The data was analysed, and a report was written to the National Blood Transfusion Service. Based on this report, the decision was taken to make the HEV screening mandatory. Since September 2020, all HEV RNA positive are confirmed molecularly, but the molecular and serological investigation of possible of X-1 and X+1 samples has not been done routinely. We are planning a follow-up manuscript describing the story of the HEV RNA screening in Switzerland starting with the HEV IgG studies we conducted in 2014-2016; then the introduction in 2018 of HEV RNA screening, to decision continue the screening mandatorily in 2022, to the experience collected to the present day. We thus would like to restrict this current manuscript to the initial study period where all the extensive data has been collected and analysed.

  1. The screening strategy was based on minipools of 12-24 samples (16-24 donations with the Roche Cobas assay and 12-15 with the Grifols Procleix assay). What is the rationale of these differences in the number of samples tested for each assay?

Switzerland has a federal system of 11 independent blood transfusion services. They are allowed to use whatever assay they wish for the NAT screening, provided they achieve the requested 450 IU/ml HEV detection limit in the individual donation. This is the reason for the different assay and pooling strategies. The following detection limits were achieved as stated in the materials and methods section: Roche cobas HEV assay 16-pools; 298 IU/ml, 24-pools; 446 IU/ml, Grifols Procleix HEV assay 12-pools;96 IU/ml and 15-pools; 120 IU/ml. Thus, the maximum and minimum 95% detection limits were 446 IU/ml and 96 IU/ml respectively.

  1. Due to several limits of detection for HEV RNA screening (limit of detection of each assay, size of the minipool) the number of positive samples and the number of donations tested should be indicated for each screening strategy. This is important for comparisons with other studies performed in Europe. These data are also relevant for a better estimation of HEV incidence in Switzerland.

This information has been added to the results section and a supplementary table added to the end of the manuscript.

Table S2: Comparison of the HEV RNA Assays, minipool sizes and HEV RNA prevalence stratified by testing centre

Testing centre

HEV RNA Assay

Minipool size

Number donations

Number HEV RNA positive

HEV RNA prevalence

Aarau

Grifols Procleix

15

42,761

10

1:4,276

Lugano

Grifols Procleix

12

20,082

4

1:5,020

Berne

Roche cobas

16

287,450

74

1:3,884

La-Chaux-de-Fonds

Roche cobas

24

35,302

4

1:8,825

Zurich

Roche cobas

24

152,141

33

1:4,610

  1. Two donors were weak HEV IgG positive in the X-1 donation indicating a reinfection. What is the anti-HEV IgG concentration in WHO units/ml?

The HEV IgG assay used throughout the study was the Wantai HEV IgG Asssy. The results are not expressed in WHO units /ml but as the ratio S/CO. Those ratios > 1.0 are considered positive. The two positive X-1 samples had in repeated tests ratios just above this cut-off (both S/CO = 1.1). They are thus weak but clearly positive. These blood donors thus have had previous contact with HEV but were not protected from reinfection.

  1. Discussion, lines 310-317: these sentences should be rewritten because the key point is the infectious dose as discussed on line 341.

As suggested by the reviewer, these sentences have been modified to include the key point that the infectious dose is paramount to the risk of transmission.

  1. Minor points

- Introduction: ICTV classification has changed (Purdy J Gen Virol 2022). HEV is now Paslahepevirus balayani.

This reference has been included and the text amended accordingly.

- Introduction: an update of the screening strategy adopted for each country (universal or selective screening, individual testing or minipool testing/size of the pools) is needed. For instance, an universal screening has replaced a selective screening in France.

Unfortunately, we have no current correct data on which country has implemented which strategies at what time. We have however, amended the text to include France for universal donor screening as requested by the reviewer.

- Discussion, line 294: the predominance of male cases is not explained by more frequent acute clinical disease because HEV RNA screening is systematic in blood donors.

We agree that the predominance of male HEV RNA positive cases is not seen in more frequent acute clinical disease. We are however, investigating in this manuscript asymptomatic blood donors and there is a clear predominance of male HEV RNA positive cases. As is written in the original text, we try and suggest possible causes. A direct proof is of course not available. We would thus prefer to leave the text as originally written.

Reviewer 2 Report

Comments and Suggestions for Authors

I have read with great interest the manuscript by Niederhauser et al. They present the data on the Prevalence of acute hepatitis E virus infections in Swiss blood donors between 2018-2020.

.The manuscript is well written and very original.

I have some minor comments to improve the clarity of their study:

-The abstract appers very long

- In the abstract the "X donation" is not clear

- France has introduced the screening for HEV RNA in all blood onors in MArch 2023

- They are 4 not confirmed blood pools: how was made the confimation?

Did they suspected molecular contamination of the instrument?

- Line 214: can you give the IgG ration of the 2 donors

- Line 221: what was the HEV RNA concentration and HEV genotype of the donor who did not seroconverted ?

- Table 3: please add the median HEV RNA concentration per genotype +

- Table 4 Can you tell us if the 10 donors that were HEV IgG positive only had high IgG concentrati

One donor was igM positive only at X+1 donation: wath was the delay between the 2 donations?

Line 327: Tipo: realstar==> Grifols

It may be interesting also to comment the study by Ushiro-Lumb (2023 Transplant Int) on organ donor screening line 340

Author Response

HEV article 2024

Answers to reviewers’ comments.

Reviewer 2

I have read with great interest the manuscript by Niederhauser et al. They present the data on the Prevalence of acute hepatitis E virus infections in Swiss blood donors between 2018-2020.

The manuscript is well written and very original.

I have some minor comments to improve the clarity of their study:

The abstract appears very long.

We have shortened the summary as suggested, without deleting the major points of the manuscript.

- In the abstract the "X donation" is not clear

The text has been amended to “individual index donation (X)” to clarify the definition, as requested.

- France has introduced the screening for HEV RNA in all blood donors in March 2023

France has been included in universal donor screening, as was also requested by reviewer 1.

- There are 4 not confirmed blood pools: how was made the confirmation?

None of these initial positive screening samples were confirmed with the quantitative HEV PCR assay (RealStar HEV RT-PCR assay) conducted at the reference centre in Bern, which has a 5 IU/ml 95% detection limit. From two of these donors, we received a follow-up sample (X+1) several months after the index donation (X). A seroconversion (HEV IgG and IgM negative) was not detected in either of these samples. A X+1 sample from the other two donors was unfortunately not obtained. It was thus assumed that all these screening HEV RNA positive results were false positive. This explanation has been added to the results section.

Did they suspect molecular contamination of the instrument?

These not confirmed donations were collected in a single regional transfusion centre over a short period of time. They have since not had unconfirmed HEV RNA positive donations. A contamination of the instrument can thus not be ruled out.

- Line 214: can you give the IgG ratio of the 2 donors

The HEV IgG assay used throughout the study was the Wantai HEV IgG Asssy. The results are not expressed in WHO units /ml but as the ratio S/CO. Those ratios > 1.0 are considered positive. The two positive X-1 samples had in repeated tests ratios just above this cut-off (both S/CO = 1.1). They are thus weak but clearly positive. These blood donors thus have had previous contact with HEV but were not protected from reinfection. This information has been added to the results section.

- Line 221: what was the HEV RNA concentration and HEV genotype of the donor who did not seroconvert?

The viral load of the HEV RNA positive index donation which did not seroconvert in the X+1 and X+2 donations was 221 IU/ml. The HEV subtype was HEV 3c, clade 3abchij. This information has been added the results section.

- Table 3: please add the median HEV RNA concentration per genotype +.

The mean HEV RNA concentration for the genotypes for those genotypes we have greater than 5 cases are as follows. This information has been mentioned in the results section and an extra table added to the supplementary material:

Table S3: The HEV RNA concentration stratified by HEV subgenotype detected.

HEV 3c
clade 3abchij

HEV 3h3
clade 3abchij

HEV 3f1
clade 3efg

HEV 3ra

Mean VL

110,596 IU/ml

71,789 IU/ml

69,883 IU/ml

968 IU/ml

Median VL

1,700 IU/ml

947 IU/ml

4,118 IU/ml

230 IU/ml

Geometric mean VL

1,692 IU/ml

1,223 IU/ml

3,095 IU/ml

334 IU/ml

Number cases

40

42

10

5

The only viral load (VL) mean between the genotypes which appears significantly different appears to be the HEV 3ra. As we do not know the exact primer binding sites of the screening and confirmatory HEV RNA assays it is impossible to determine if this rabbit genotype has generally lower viral loads or if this is a technical problem of the assays.

- Table 4 Can you tell us if the 10 donors that were HEV IgG positive only had high IgG concentration.

Five of the 10 had very high IgG concentrations (S/CO >15), the remaining 5 had S/CO values between 1.1 to 3. This information has been placed as an addendum to Table 4.

One donor was IgM positive only at X+1 donation: with was the delay between the 2 donations?

This X+1 sample was IgG negative, but had a S/CO value 0.8, thus just not positive as the cut-off is 1. The IgM value was S/CO 1.6, thus weak positive. It was collected 7 months after the index donation (X). This information has also been placed as an addendum to Table 4.

Line 327: Tipo: realstar==> Grifols

Changed as correctly suggested.

It may be interesting also to comment the study by Ushiro-Lumb (2023 Transplant Int) on organ donor screening line 340

A discussion of this interesting article has been added to the discussion section and the references updated. We thank the reviewer for pointing out this article.

Reviewer 3 Report

Comments and Suggestions for Authors

The authors investigated the incidence of hepatitis E virus (HEV) infection among Swiss blood donors, determined the respective viral loads of these blood donations, identified HEV genotypes and subgenotypes, and established the timing of HEV seroconversions among the blood donors.

The data obtained in this study provide compelling evidence and are instrumental in informing decisions regarding the implementation of mandatory HEV RNA universal screening for all Swiss blood donations in minipools. However, there are some minor concerns in this manuscript that necessitate attention, as delineated below.

Comments:

1.    The taxonomic classification of the family Hepeviridae has recently been revised, dividing the virus family into two subfamilies: Orthohepevirinae with four genera (Paslahepevirus, Avihepevirus, Rocahepevirus, and Chirohepevirus) and Parahepevirinae with one genus (Piscihepevirus) (Purdy et al., J Gen Virol. 2022; 103:001778; https://ictv.global/report/chapter/hepeviridae). Therefore, the descriptions regarding HEV nomenclature should be updated (Line 45).

2.    Lines 158-159: "6’029 to 6’521" should be amended to "6,029 to 6,521".

3.    Line 160: "5’752 to 6’340" should be corrected to "5,752 to 6,340".

4.    Line 165: "6’029 to 6’340" should be revised to "6,029 to 6,340".

5.    Line 167: "1’000" should be rectified to "1,000".

6.    Table 1: "85’802" should be adjusted to "85,802".

7.    Table 4: The median time between X-1 and X should be included in category B.

8.    Figure 2: In the title, the period "." after 30 should be removed.

9.    Figure 3: Bootstrap values should be indicated at each node.

Author Response

HEV article 2024

Answers to reviewers’ comments.

Reviewer 3

The authors investigated the incidence of hepatitis E virus (HEV) infection among Swiss blood donors, determined the respective viral loads of these blood donations, identified HEV genotypes and subgenotypes, and established the timing of HEV seroconversions among the blood donors.

The data obtained in this study provide compelling evidence and are instrumental in informing decisions regarding the implementation of mandatory HEV RNA universal screening for all Swiss blood donations in minipools. However, there are some minor concerns in this manuscript that necessitate attention, as delineated below.

Comments:

  1. The taxonomic classification of the family Hepeviridae has recently been revised, dividing the virus family into two subfamilies: Orthohepevirinae with four genera (Paslahepevirus, Avihepevirus, Rocahepevirus, and Chirohepevirus) and Parahepevirinae with one genus (Piscihepevirus) (Purdy et al., J Gen Virol. 2022; 103:001778; https://ictv.global/report/chapter/hepeviridae). Therefore, the descriptions regarding HEV nomenclature should be updated (Line 45).

This has been corrected and the reference list updated.

  1. Lines 158-159: "6’029 to 6’521" should be amended to "6,029 to 6,521".

Amended as suggested.

  1. Line 160: "5’752 to 6’340" should be corrected to "5,752 to 6,340".

Amended as suggested.

  1. Line 165: "6’029 to 6’340" should be revised to "6,029 to 6,340".

Amended as suggested.

  1. Line 167: "1’000" should be rectified to "1,000".

Amended as suggested.

  1. Table 1: "85’802" should be adjusted to "85,802".

Amended as suggested.

  1. Table 4: The median time between X-1 and X should be included in category B.

The median time between X-1 and X is 147 days. This has been included in Table 4 B.

  1. Figure 2: In the title, the period "." after 30 should be removed.

Amended as suggested.

  1. Figure 3: Bootstrap values should be indicated at each node.

The adding of bootstraps to the circular phylogenetic analysis portrayal has proved unreadable as it is too small. We would prefer to add a second linear phylogenetic analysis drawing as a supplementary figure:

Figure S1: Linear phylogenetic analysis of the HEV variants from Swiss blood donors, October 2018 - September 2020 with bootstrap values

Round 2

Reviewer 1 Report

Comments and Suggestions for Authors

The answers are acceptable